



**Contrasting sources and processes of particulate species in haze days with low and**
**high relative humidity in wintertime Beijing**
Ru-Jin Huang[1], Yao He[1], Jing Duan[1], Yongjie Li[2], Qi Chen[3], Yan Zheng[3], Yang Chen[4], Weiwei
Hu[5], Chunshui Lin[1], Haiyan Ni[1], Wenting Dai[1], Junji Cao[1], Yunfei Wu[6], Renjian Zhang[6], Wei
Xu[1,7], Jurgita Ovadnevaite[7], Darius Ceburnis[7], Thorsten Hoffmann[8], Colin D. O'Dowd[7]
[1]State Key Laboratory of Loess and Quaternary Geology, Center for Excellence in
Quaternary Science and Global Change, and Key Laboratory of Aerosol Chemistry and
Physics, Institute of Earth and Environment, Chinese Academy of Sciences, Xi'an 710061,
China
[2]Department of Civil and Environmental Engineering, Faculty of Science and Technology,
University of Macau, Taipa, Macau, China
[3]State Key Joint Laboratory of Environmental Simulation and Pollution Control, College
of Environmental Sciences and Engineering, Peking University, Beijing 100871, China
[4]Chongqing Institute of Green and Intelligent Technology, Chinese Academy of Sciences,
Chongqing 400714, China
[5]State Key Laboratory of Organic Geochemistry and Guangdong Key Laboratory of
Environmental Protection and Resources Utilization, Guangzhou Institute of
Geochemistry, Chinese Academy of Sciences, Guangzhou 510640, China
[6]RCE-TEA, Institute of Atmospheric Physics, Chinese Academy of Sciences, Beijing
100029, China
[7]School of Physics and Centre for Climate and Air Pollution Studies, Ryan Institute,
National University of Ireland Galway, University Road, Galway H91CF50, Ireland
[8]Institute of Inorganic and Analytical Chemistry, Johannes Gutenberg University of
Mainz, Duesbergweg 10-14, 55128 Mainz, Germany
*Correspondence to*: Ru-Jin Huang (rujin.huang@ieecas.cn)
**Abstract**
Although there are many studies of particulate matter (PM) pollution in Beijing, the
sources and processes of secondary PM species during haze periods remain unclear.
Limited studies have investigated the PM formation in highly-polluted environments
under low and high relative humidity (RH) conditions. Herein, we present a systematic
comparison of species in submicron particles ($PM_1$) in wintertime Beijing (29 December
2014 to 28 February 2015) for clean periods and pollution periods under low and high
RH conditions. $PM_1$ species were measured with an aerosol chemical species monitor
(ACSM) and an aethalometer. Sources and processes for organic aerosol (OA) were
resolved by positive matrix factorization (PMF) with multilinear engine 2 (ME-2). The
comparisons for clean, low-RH pollution, and high-RH pollution periods are made from
three different aspects, namely (a) mass concentration, (b) mass fraction, and (c) growth
rate in diurnal profiles. OA is the dominant component of $PM_1$ with an average mass
concentration of 56.7 μg m$^{-3}$ (46%) during high-RH pollution and 67.7 μg m$^{-3}$ (54%)
during low-RH pollution periods. Sulfate had higher concentration and mass fraction





during high-RH pollution periods, while nitrate had higher concentration and mass
fraction during low-RH pollution periods. The diurnal variations of nitrate and
oxygenated organic aerosol (OOA) showed a daytime increase of their concentrations
during all three types of periods. Nitrate had similar growth rates during low-RH (0.40 µg
$m^{-3}$ $h^{-1}$) and high-RH (0.55 µg $m^{-3}$ $h^{-1}$) pollution periods. OOA had a higher growth rate
during low-RH pollution periods (1.0 µg $m^{-3}$ $h^{-1}$) than during high-RH pollution periods
(0.40 µg $m^{-3}$ $h^{-1}$). In contrast, sulfate had a decreasing trend during low-RH pollution
periods, while it increased significantly with a growth rate of 0.81 µg $m^{-3}$ $h^{-1}$ during high-
RH pollution periods. These distinctions in mass concentrations, mass fractions, and
daytime growth rates may be explained by the difference in the formation processes,
affected by meteorological conditions. In particular, photochemical oxidation and
aqueous-phase processes may both produce sulfate and nitrate. The relative importance
of the two pathways, however, differs under different meteorological conditions.
Additional OOA formation under high-RH (>70%) conditions suggests aqueous-related
formation pathways. This study provides a general picture of the haze formation in Beijing
under different meteorological conditions.
**1 Introduction**
Air pollution is a serious environmental problem in China, particularly in the North China
Plain (NCP) in winter, affecting air quality and human health. Beijing is one of the most
polluted megacities in the NCP, with an annual mean concentration of $PM_{2.5}$ being 86 and
51 µg $m^{-3}$ in 2014 and 2018, respectively (http://sthjj.beijing.gov.cn/), which significantly
exceeded the Chinese National Ambient Air Quality Standard (annual average of 35 µg $m^{-3}$).
Fine PM pollution in polluted urban environments is complex and is typically
associated with enhanced primary emissions from multiple sources, strong secondary
aerosol formation, and stagnant weather conditions (Sun et al., 2011; 2013; 2016; Huang
et al., 2014; Hu et al., 2016; An et al., 2019). Regional transport of air pollutants from
urbanized and industrialized areas has an important contribution to fine PM pollution in
the NCP region. For example, severe fine PM pollution in Beijing during winter often
happened when prevailing air masses were from the south (Sun et al., 2016).
Organic aerosol (OA) is the major constituent of fine PM and is much less understood
compared to inorganic aerosol in terms of their chemical nature and sources (Hallquist et
al., 2009; Shrivastava et al., 2017). OA is composed of a wide variety of organic species
from different sources, and its emission sources and atmospheric processes are not well
understood so far, especially in those regions with high fine PM pollution. OA is either
directly emitted to the atmosphere (primary organic aerosol, POA) or formed in the
atmosphere (secondary organic aerosol, SOA). Therefore, it is essential to identify and
quantify the major emission sources and understand the formation processes of OA.
The Aerodyne aerosol chemical speciation monitor (ACSM) with quadrupole (Q) or time-
of-flight (TOF) mass analyzer is capable of real-time determination of non-refractory
components in submicron aerosol ($NR-PM_1$), overcoming the limitation of filter





82 measurements such as limited time resolution or measurement artifacts (Ng et al., 2011a;
83 Froehlich et al., 2013). ACSM has been widely used for fine PM studies in many sites in
84 China including Beijing, Nanjing, Shijiazhuang, and Baoji (Sun et al., 2014; Wang et al.,
85 2017; Zhang et al., 2017; Huang et al., 2019). By applying positive matrix factorization
86 (PMF, Paatero et al., 1993) or multilinear engine (ME-2, Canonaco et al., 2013) solver to
87 the ACSM data, main OA sources can be identified. Those sources include hydrocarbon-
88 like OA (HOA), biomass burning OA (BBOA), cooking OA (COA), coal combustion OA (CCOA)
89 and oxygenated OA (OOA). OOA can further be resolved into semi-volatile OOA (SV-OOA)
90 and low-volatility OOA (LV-OOA) by volatility, or more-oxidized OOA (MO-OOA) and less-
91 oxidized OOA (LO-OOA) by oxidation state. MO-OOA and LO-OOA together were found to
92 contribute 61% of OA in Beijing during summer in 2011 (Sun et al., 2012), while POA was
93 found to be more important during winter of the same year (Sun et al., 2013). However,
94 many recent studies show large contributions of SOA in wintertime Beijing (Huang et al.,
95 2014; Hu et al., 2016; Xu et al., 2018) and CCOA is often found to be a large fraction of POA
96 during wintertime pollution days in Beijing (Sun et al., 2016b; Wang et al., 2015; Elser et
97 al., 2016). The discrepancies in SOA contribution in different measurement periods reflect
98 the difference in atmospheric and meteorological conditions, e.g., atmospheric oxidative
99 capacity and relative humidity (RH) (Sun et al., 2013; Xu et al., 2017; Wu et al., 2018; Song
100 et al., 2019).

101 In this study, we present measurement results at an urban site in Beijing during the winter
102 of 2014-2015. The chemical nature of NR-PM$_1$, sources of OA, formation processing of
103 secondary aerosol in different episodes, and particularly the effects of RH on secondary
104 aerosol formation are discussed.

105 **2 Methods**

106 **2.1 Site description and instrumentation**

107 The online measurements were conducted on the rooftop of a building (about 20 m above
108 the ground level) at the campus of the National Centre for Nanoscience and Technology
109 (40.00° N, 116.38° E) from 29 December 2014 to 28 February 2015. The observation site
110 is between the 4$^{th}$ and 5$^{th}$ ring roads in the northwest of Beijing and is surrounded by a
111 residential area.

112 A Q-ACSM was deployed for the mass concentration measurements of NR-PM$_1$ species,
113 and the detailed operation principles can be found in Ng et al. (2011a). Briefly, ambient
114 air was pumped through a 3/8 in stainless steel tube at a flow rate of 3 L min$^{-1}$, of which
115 85 mL min$^{-1}$ was sampled into the Q-ACSM. In order to remove coarse particles, an URG
116 cyclone (URG-2000-30ED, size cut-off 2.5 μm) was installed in front of the inlet. Because
117 particle bounce can affect collection efficiency (CE), to reduce this uncertainty and to dry
118 the particles, a Nafion dryer (MD-110-48S; Perma Pure, Inc., Lakewood, NJ, USA) was
119 installed after the URG cyclone. An aerodynamic lens was used to focus the submicron
120 particles into a narrow beam, the particles beam then impinged on a heated tungsten



surface (about 600 °C) to evaporate, impacted by 70-eV electron to ionize, and then
detected by a quadrupole mass spectrometer. During this study, the scan rate of Q-ACSM
was at 200 ms amu$^{-1}$ from m/z 10 to 150 and the time resolution was 30 min. To determine
the response factor (RF), a differential mobility analyzer (DMA, TSI model 3080) and a
condensation particle counter (CPC, TSI model 3772) were used to select and count the
monodisperse 350-nm ammonium nitrate ($NH_4NO_3$) particles, respectively. The mass of
$NH_4NO_3$ particles was calculated with known particle size and number concentrations.
This calculated mass concentration was compared to the RF of the Q-ACSM, resulting in
the ionization efficiency (IE) value (Ng et al., 2011a).
The gaseous species including $O_3$ and $NO_x$ were measured by a Thermo Scientific Model
49i ozone analyzer and a Thermo Scientific Model 42i $NO–NO_2–NO_x$ analyzer, respectively.
The $NH_3$ concentrations were measured by an $NH_3$ analyzer (Picarro G2103). The
concentrations of black carbon (BC) was determined by an aethalometer (Model AE-33,
Magee Scientific) with a time resolution of 1 min. In brief, light attenuation at seven
different wavelengths was recorded for particle-laden filter spots, and BC concentration
was retrieved based on the light attenuation at 880 nm. An automatic weather station
(MAWS201, Vaisala, Vantaa, Finland) was used to measure the meteorological parameters
including temperature, pressure, relatively humidity and visibility, and a wind sensor
(Vaisala Model QMW101-M2) was used to measure the wind speed and wind direction.
**2.2 Data analysis**
**2.2.1 ACSM data analysis**
The standard Q-ACSM data analysis software (v.1.5.3.5) written in Igor Pro (WaveMetrics,
Inc., OR, USA) was used to calculate the mass concentrations for different species in NR-
$PM_1$. Default relative ionization efficiencies (RIE) were used for organics (1.4), nitrate (1.1)
and chloride (1.4), respectively (Ng et al., 2011a). RIE of 5.8 for ammonium and 1.2 for
sulfate were determined by the IE calibrations of ammonium nitrate and ammonium
sulfate. Meanwhile, data were corrected for the particle collection efficiency (CE), due to
particle bounce on the vaporizer. CE can be affected by relative humidity, mass fraction of
ammonium nitrate and particle acidity. In our measurement, the particles were generally
neutral and dried before sampling into the ACSM. CE was calculated as $CE_{dry}$ = max (0.45,
0.0833 + 0.9167 × ANMF), where ANMF refers to the ammonium nitrate fraction in NR-
$PM_1$(Middlebrook et al. 2012).
**2.2.2 OA source apportionment**
The receptor model PMF using a multilinear engine (ME-2) was used to identify and
quantify the OA sources. PMF is a bilinear receptor model used to describe the variability
of a multivariate dataset, X, as the linear combination of a set of constant factor profiles, F,
and their corresponding time series G, as expressed in equation 1.



$X = GF + E$      (1)
where X is the measured OA mass spectra consisting of $i$ rows and $j$ columns, and E is the
model residuals. The PMF uses a least squares method to minimize the object function $Q$,
defined as the sum of the squared residuals ($e_{ij}$) weighted by their respective uncertainties
($\sigma_{ij}$).
$Q = \sum_{i=1}^{m} \sum_{j=1}^{n} (e_{ij}/\sigma_{ij})^2$      (2)
Unconstrained PMF analyses of OA data suffer from rotational ambiguity when sources
show similar profiles and temporal covariation (Canonaco et al., 2013; Huang et al., 2019).
However, by introducing *a priori* information as additional model input and constraining
one or more output factor profiles to a predetermined range, ME-2 can overcome such
difficulties and provide more environmentally meaningful solutions. When an element of
a factor profile ($f_j$ , where $j$ refers to the *m/z*) is constrained with a certain *a* value (a), the
following conditions need to be fulfilled:
$f_{j,solution} = f_j \pm a \times f_j$      (3)
The *a* value can vary between 0 and 1, which is the extent to which the output profiles can
vary from the model inputs. The data analysis were conducted using the source finder
(SoFi, Canonaco et al., 2013) tool version 4.9 for Igor Pro. Due to rotational ambiguity,
there was no mathematically unique solution. Therefore, criteria including chemical
fingerprint of the factor profiles, correlations with external tracers, and diurnal cycles
were used for the factor identification and interpretation (Ulbrich et al., 2009; Huang et
al., 2014, Elser et al., 2016 ).
**2.2.3 Aerosol liquid water content**
NR-PM$_1$ inorganic species, NH$_3$ concentrations and meteorological parameters including
temperature and RH were used to calculate the aerosol liquid water content (ALWC)
based on the ISORROPIA-II model (Fountoukis and Nenes, 2007). Here we ran the
ISORROPIA-II in "forward" mode and the particles were assumed to be deliquescent, i.e.,
in metastable mode (Hennigan et al., 2015). The thermodynamic equilibrium of the $NH_4^+$–
$SO_4^{2-}$–$NO_3^-$–$Cl^-$–$H_2O$ system was then modeled and ALWC was calculated.
**3 Results and discussion**
**3.1 Temporal variations and mass fractions of PM$_1$ species**
Fig. 1 shows the time series of mass concentrations of OA, $SO_4^{2-}$, $NO_3^-$, $NH_4^+$, $Cl^-$, and BC, as
well as the meteorological parameters. The average mass concentration of PM$_1$ during the
entire measurement period was 73.8 µg m$^{-3}$, similar to those observed in Beijing in winter
2011 (66.8 µg m$^{-3}$, Sun et al., 2013) and winter 2013 (64 µg m$^{-3}$, Sun et al., 2016). The





lowest daily average concentration was 5.2 µg m$^{-3}$ on 31 December, while the highest was
210.1 µg m$^{-3}$ on 15 January, with a difference of a factor of ∼40. OA (52%) was the most
abundant component of PM$_1$, irrespective of the meteorological conditions, followed by
nitrate (14%) and sulfate (11%). The weather conditions during the measurement period
were characterized by drastic changes in wind speed, wind direction, RH and temperature,
providing a unique setting to investigate the influence of meteorological conditions on PM
species. As such, the entire measurement period can be divided into the clean period (PM$_1$
<20 µg m$^{-3}$) and the pollution period (PM$_1$ >100 µg m$^{-3}$). South/southeasterly wind
directions with low speed (average, 0.9 -1.0 m s$^{-1}$) were typical for the pollution period,
while north/northwesterly with high speed (average, 2.5 m s$^{-1}$) for the clean period (Table
202  1).

To investigate the effects of RH on PM pollution formation, we further divided the
pollution period into two categories, the low-RH pollution days (RH <50%) and the high-
RH pollution days (RH >50%). The diurnal variations of mass concentrations and fractions
of different chemical species during clean days, low-RH pollution days and high-RH
pollution days are shown in Fig. 2. The mass fractional variations were flatter during low-
RH and high-RH pollution days than during clean days, likely due to the accumulation of
pollutants during stagnant weather conditions in pollution days. During clean days,
secondary inorganic aerosol showed generally increasing trends from 06:00 to 20:00 local
time (LT), despite the development of the boundary layer height during the day. The
growth rate of nitrate mass (0.21 µg m$^{-3}$ h$^{-1}$) was higher than that of sulfate (0.04 µg m$^{-3}$ h$^{-1}$)
and ammonium (0.10 µg m$^{-3}$ h$^{-1}$), indicating that formation of nitrate was perhaps faster
than that of sulfate and ammonium during clean days. During low-RH pollution days,
nitrate increased from 06:00 to 20:00 LT, with a growth rate of 0.40 µg m$^{-3}$ h$^{-1}$, which was
two times higher than that during clean days. On the contrary, sulfate concentrations
increased from 06:00 to 10:00 LT, then started decreasing and reached the minimum at
14:00 LT, possibly due to the increase of the boundary layer height during the day, which
outweighed the production of sulfate. Associated with both sulfate and nitrate,
ammonium showed a minor increase from 06:00 to 20:00 LT with a mass growth rate of
0.18 µg m$^{-3}$ h$^{-1}$. This phenomenon suggested that the low-RH condition was favorable for
nitrate formation but not for sulfate formation under polluted conditions. In contrast,
obvious increases of secondary inorganic species from 8:00 to 16:00 LT were observed
during high-RH pollution days, with growth rates of 0.81 µg m$^{-3}$ h$^{-1}$, 0.55 µg m$^{-3}$ h$^{-1}$ and 0.46
µg m$^{-3}$ h$^{-1}$ for sulfate, nitrate and ammonium, respectively. These mass growth rates
increased correspondingly by about 20, 2.6 and 4.6 times compared to those during clean
days. Note that nitrate growth rate in high-RH pollution days (0.55 µg m$^{-3}$ h$^{-1}$) was still
slightly higher than that in low-RH pollution days (0.40 µg m$^{-3}$ h$^{-1}$), indicating that nitrate
production is still efficient when RH is high, although not as much higher compared to
sulfate. Measurements of sulfate oxygen isotopes suggest that the largely enhanced
formation of sulfate is associated with efficient aqueous-phase reactions during high-RH
pollution days (Shao et al., 2019).
**3.2 Sources and diurnal variations of OA**



Source apportionment was performed on the OA data. Three to seven factors were
examined using an unconstrained PMF model, and the factors were qualitatively identified
based on their mass spectral profiles and correlation with external data. We found that a
solution of five factors (i.e., HOA, COA, CCOA, BBOA, and OOA) best explains our data. For
the solutions with less than 5 factors, HOA appeared to be mixed with COA while CCOA
mixed with BBOA (Fig. S1). However, when the number of factors was increased to 6, the
OOA factor split into two OOA factors of similar time series (Fig. S2), suggesting that
further separation of the factors does not improve the interpretation of the data.
Although five factors with different profiles and temporal variations were identified by
the unconstrained PMF model, the factor profiles and time series were suboptimal,
specifically for HOA, COA, and BBOA. The diurnal pattern of HOA showed pronounced
peaks at cooking time, indicating its mixing with COA. The fractional contribution of m/z
($f_{60}$, typically related to the fragmentation of anhydrous sugars) in HOA (0.008) was
higher than the average value reported from multiple ambient datasets (0.002, Ng et al.,
2011). To reduce the mixing between factors, the reference HOA mass spectral profile,
characterized by a small $f_{60}$ (Wang et al., 2017), and the BBOA mass spectral profile,
derived from Beijing wintertime measurements (Elser et al., 2016), were constrained
using ME-2. For the COA mass spectral profile that was derived from our unconstrained
PMF analysis, a-value of 0 was used. Meanwhile, for HOA and BBOA, the $a$ values were
varied systematically between 0 and 1 with an interval of 0.1 to explore the solution space.
To assess the obtained solutions, we have set thresholds for the highest acceptable $f_{60}$
value (0.006) for HOA and $f_{57}$ value (0.042) for BBOA, based on mass spectra obtained at
multiple sites (mean ± 2σ, Ng et al., 2011). Only solutions that conform to both criteria
were selected and the final solution was the average of those selected reasonable
solutions (Fig. S3).
The final OA factors resolved by ME-2 include four POA (i.e., HOA, COA, BBOA and CCOA),
and one SOA (i.e., OOA) factors, on average accounting for 14%, 14%, 10%, 32% and 31%
of OA mass concentration, respectively. The mass spectral profiles and time series of the
resolved factors are shown in Fig. 3a and b, respectively. The diurnal patterns of these
factors are presented in Fig. 4. The HOA spectrum is similar to those derived from other
studies in Beijing (Hu et al., 2016; Sun et al., 2014; 2016) and Pittsburgh (Ulbrich et al.,
2016), and also resembles the source profile from diesel exhausts (Canagaratna et al.,
2004). A strong correlation between the time series of HOA and BC was observed
($R^2$=0.84). The diurnal cycle of HOA was similar to those observed in other studies in
Beijing (Sun et al., 2011; 2013; 2014), showing higher mass concentrations during the
night than during the day, due to enhanced traffic emissions from heavy duty vehicles and
diesel trucks that are allowed to enter the inner city during the night.
Similar to HOA, the mass spectrum of COA also displayed high signals in odd fragments,
while the $m/z$ 55/57 ratio (1.45) and $m/z$ 41/43 ratio (1.6) were significantly higher
compared to those of the HOA factor profile ($m/z$ 55/57=0.65, $m/z$ 41/43=0.88). The COA
profile is similar to those resolved in previous studies in Beijing (Elser et al., 2016; Sun et





al., 2016), Paris (Crippa et al., 2013) and Zurich (Dey et al., 2004). The $R^2$ between COA
and $m/z$ 55 time series was 0.73. The diurnal cycle of COA showed two prominent peaks
during lunch (12:00-13:00 LT) and dinner (18:00-19:00 LT) times, and the peak in the
evening was more pronounced than that at noon, consistent with a previous study in
Beijing (Sun et al., 2016). Furthermore, the diurnal variation of COA was more obvious
with much clear noon and evening peaks during clean days than during low-RH and high-
RH pollution days, likely because the stagnant meteorological conditions during pollution
days facilitated the accumulation of pollutants and thus weakened the diurnal fluctuation.
The BBOA mass spectrum showed a similar pattern as that extracted from Crippa et al.
(2014), with pronounced peaks at $m/z$ 60 and 73, two distinct markers of biomass
burning emissions (Lanz et al., 2007). BBOA also showed similar time series with a high
signal at $m/z$ 60 ($R^2$=0.74). The diurnal cycle of BBOA showed a slight increase during the
night (18:00-24:00 LT), corresponding to nighttime burning for residential heating in
clean days, while this diurnal cycle became much flat during low-RH and high-RH
pollution days, likely due to the stagnant meteorological conditions during pollution days.
On average, BBOA contributed 10% of the total OA, much less than that of CCOA (32%),
consistent with previous results in Beijing (Elser et al., 2016).
The profile of CCOA showed a moderate correlation with that resolved in Beijing in winter
2014 (Elser et al., 2016). Similar to previous studies, signals related to unsaturated
hydrocarbons, especially those at $m/z$ 77, 91 and 115, contributed significantly to the total
CCOA signal. In addition, there was a strong correlation between CCOA and Cl⁻ ($R^2$=0.82),
which was considered as a marker mainly from coal combustion emissions. The mass
concentration and mass fraction of CCOA were both significantly higher at night than
those during the day, which was observed both in clean days and pollution days. The
diurnal pattern suggests much stronger emissions from coal combustion at night, a
situation further deteriorated by a shallower boundary layer at night.
One secondary OA factor, namely OOA, was also resolved, characterized by an important
contribution at $m/z$ 44. The profile of OOA is also similar to those resolved in Ng et al.
(2011) and Sun et al. (2013). OOA is correlated well with nitrate ($R^2$=0.89).   and the
diurnal cycle of OOA shows an increase from about 6:00 to 20:00 LT, indicating the
contribution from photochemical production. Note that the growth rate of OOA during
low-RH pollution days (1.0 $\mu g\,m^{-3}\,h^{-1}$) was higher than that during high-RH pollution days
(0.40 $\mu g\,m^{-3}\,h^{-1}$) and clean days (0.35 $\mu g\,m^{-3}\,h^{-1}$) (Fig. 4).

### 3.3 Chemically resolved PM pollution

Fig. 5 shows the mass fraction of $PM_1$ and OA during clean, low-RH and high-RH pollution
periods. OA was the dominant component in $PM_1$, with an average concentration
increasing from 10.9 $\mu g\,m^{-3}$ during clean periods to 56.7 $\mu g\,m^{-3}$ during high-RH pollution
periods and further to 67.7 $\mu g\,m^{-3}$ during low-RH pollution periods. The corresponding
mass fraction of OA was 56%, 46%, and 54%, respectively. The decrease of OA mass



fraction during pollution periods can be attributed to the increased formation of sulfate
and nitrate, as demonstrated in the above section. Specifically, nitrate increased from 11%
(2.2 µg m$^{-3}$) during clean periods to 14% (17.2 µg m$^{-3}$) during high-RH pollution periods
and to 15% (18.8 µg m$^{-3}$) during low-RH pollution periods, while sulfate increased from
10% (2.0 µg m$^{-3}$) during clean periods to 17% (20.9 µg m$^{-3}$) during high-RH pollution
periods but decreased back to as low as 7% (8.8 µg m$^{-3}$) during low-RH pollution periods.
The increased formation of nitrate from clean to pollution periods, especially during low-
RH pollution periods, is likely due to enhanced photochemical production, as discussed in
Lu et al. (2019) which shows fast photochemistry during wintertime haze events in
Beijing. Specifically, the atmospheric oxidation proxy ($O_x=O_3+NO_2$) increased from 39.2
ppb during clean periods to 47.8 ppb during high-RH pollution periods, and up to as high
as 59.8 ppb during low-RH pollution periods. Meanwhile, the precursor gas for nitrate,
$NO_2$, increased accordingly from 16.7 ppb during clean periods to 64.3 ppb during high-
RH pollution periods and to 103.0 ppb during low-RH pollution periods. The averaged
PM$_1$ concentrations during high-RH (123.2 µg m$^{-3}$) and low-RH (125.4 µg m$^{-3}$) pollution
periods were very similar, but a distinct difference lies in the sulfate and nitrate fractions
in these two types of pollution periods. We observed a much larger contribution from
nitrate during low-RH pollution periods than during high-RH pollution periods, which
may be due to enhanced photochemical formation and also contributions of $N_2O_5$ uptake,
and a much larger contribution from sulfate during high-RH pollution periods than during
low-RH pollution periods because of enhanced formation from aqueous-phase processes.
In terms of OA sources, CCOA and OOA were the major sources irrespective of the PM$_1$
level. The mass fraction of CCOA in OA increased from 25% (2.8 µg m$^{-3}$) during clean
periods to 31% (17.6 µg m$^{-3}$) during high-RH pollution periods and to 35% (23.7 µg m$^{-3}$)
during low-RH pollution periods, indicating the important contribution of residential coal
combustion emissions. OOA also increased significantly during pollution periods, from 4.1
µg m$^{-3}$ to ~20 µg m$^{-3}$. It should be noted that the average OOA mass concentrations were
rather similar during high-RH (19.8 µg m$^{-3}$) and low-RH (18.3 µg m$^{-3}$) pollution periods.
However, the OOA mass fraction in OA during the high-RH pollution period (35%) is
higher than that during the low-RH pollution period (27%), indicating an additional
contribution of OOA from e.g., aqueous-phase oxidations during high RH condition, as
discussed below. The mass fraction of HOA in OA increased from 8% (0.8 µg m$^{-3}$) during
clean days to 13% (8.8 µg m$^{-3}$) during low-RH pollution days and further to 16% (9.1 µg
m$^{-3}$) during high-RH pollution days, suggesting an increased contribution of HOA in
pollution days. The mass fraction of HOA is similar to those measured in wintertime
Beijing in 2011(14%, Hu et al., 2016) and in 2013 (11%, Sun et al., 2016). In contrast, the
mass concentrations of COA during low-RH pollution days (8.8 µg m$^{-3}$) and high-RH
pollution days (6.8 µg m$^{-3}$) were higher than that during clean days (2.0 µg m$^{-3}$), but the
mass fraction of COA in OA during high-RH pollution days (12%) and low-RH pollution
days (13%) were lower than that during clean days (20%). A similar decrease of HOA
contribution and increase of COA contribution during clean days were also observed by
Sun et al. (2016) in wintertime Beijing in 2011. The highest contribution of BBOA was
observed during low-RH pollution days with a mass fraction of 12% (8.1 µg m$^{-3}$). The





BBOA concentration during high-RH pollution days (3.4 µg m$^{-3}$) was higher than that
during clean days (1.0 µg m$^{-3}$), but the mass fraction of BBOA in OA during high-RH
pollution days (6%) was lower than that during clean days (10%).
The chemical composition and sources of PM$_1$ under different meteorological conditions
(e.g., wind direction, wind speed and RH) in the seven pollution episodes (PM$_1$ >100 µg
m$^{-3}$) and seven clean episodes (PM$_1$ <20 µg m$^{-3}$) are shown in Fig. S4. Note that these
episodes in total accounted for 91% of the entire measurement period. The pollution
episodes were found to be associated with the air masses from south/southwest, while
clean episodes were associated with the air masses from north/northwest. Meanwhile,
the pollution episodes were generally associated with higher RH and lower wind speeds
when compared to the clean episodes. The wind speeds were approximately three times
higher in clean episodes than those in pollution episodes. For example, the lowest
concentration of PM$_1$ was 6.7 µg m$^{-3}$ in C6 period, corresponding to the highest wind speed
(4 m s$^{-1}$) and the lowest concentrations (< 20 ppb) of inorganic gaseous precursors (SO$_2$,
NH$_3$, and NO$_x$), while the highest PM$_1$ concentration of 169 µg m$^{-3}$ was found at P5,
corresponding to a much lower wind speed (<1 m s$^{-1}$). The mass concentrations of OA
increased from ~4.1-9.4 µg m$^{-3}$ during clean episodes to ~44.7-85.7 µg m$^{-3}$ during
pollution episodes. However, the contributions of OA to PM$_1$ showed a decreasing trend
from 48-59% during clean episodes to 44-57% during pollution episodes, and the
corresponding contributions of secondary inorganic species increased from 29-34%
(~2.2-5.5 µg m$^{-3}$) to 27-47% (~25.5-62.1 µg m$^{-3}$), indicating a notable production and
accumulation of secondary inorganic aerosol during haze pollution episodes. In contrast,
the mass concentration of OOA increased from ~1.4-3.9 µg m$^{-3}$ during clean episodes to
~10.0-27.6 µg m$^{-3}$ during pollution episodes, while the contribution of OOA to OA
decreased from 33-64% during clean episodes to 20-52% during pollution episodes. The
corresponding contribution of POA sources increased from 35-67% (~1.2-4.7 µg m$^{-3}$) to
38-80% (~13.9-58.7 µg m$^{-3}$), suggesting that in general the emission and accumulation of
POA sources played an important role during haze pollution in this measurement
campaign.
Comparing the pollution episodes with different RH conditions (see Fig. S4), the mass
fraction of sulfate was much higher during high-RH pollution episodes (P3, P6 and P7, 15-
21%) than during low-RH pollution episodes (P1, P2, P4 and P5, 6-8%). OOA also showed
a much higher contribution to OA during high-RH pollution events (P6, P7, 50-62%) than
during low-RH pollution events (P1-P5, 20-31%). These variations suggest the potential
importance of aqueous-phase reactions on the formation of sulfate and OOA, as discussed
above. Further comparison of high-RH and low-RH pollution episodes with similar PM
levels (e.g., P2 and P6 with PM$_1$ concentration of 98.8 µg m$^{-3}$ and 99.6µg m$^{-3}$, respectively)
shows that secondary inorganic aerosol dominated PM$_1$ at high-RH pollution episode.
Similarly, as for the high-RH and low-RH pollution episodes with similar OA levels, for
example, P6 (44.7µg m$^{-3}$) and P7 (46.3 µg m$^{-3}$), OOA dominated the particulate pollution
(62% of OA) at high-RH pollution events due to efficient formation of SOA. On the contrary,
POA had increased contributions to PM pollution at low RH and stagnant weather





conditions (from 38% of OA at high-RH pollution to 50% of OA at low-RH pollution),
consistent with previous studies in other Chinese cities (e.g., Wang et al., 2017; Huang et
al., 2019). These results indicate that meteorological conditions have important effects on
the particulate pollution.
**3.4 Formation of secondary aerosol**
The relationship between $SO_4^{2-}$ and $NO_3^-$ is investigated to elucidate the formation
processes of these two typical secondary inorganic aerosol species. The correlation
between $SO_4^{2-}$ and $NO_3^-$ was weak for the entire pollution period, because of the varied
relative contribution of different formation processes during different periods. However,
better correlations between $SO_4^{2-}$ and $NO_3^-$ were found with different slopes when the data
were divided into low-RH (RH <50%) and high-RH (RH >50%) pollution periods (Fig. 6).
During low-RH pollution periods, $NO_3^-$ and $SO_4^{2-}$ showed a good correlation ($R^2$ = 0.75)
with a ratio of 2.1, indicating a similar photochemical production process. However,
during high-RH pollution periods, the ratio of $NO_3^-$ to $SO_4^{2-}$ decreased significantly to 0.40
with a lower correlation coefficient ($R^2$ = 0.53). The degraded temporal correlation
between nitrate and sulfate suggest different formation pathway of nitrate and sulfate
during high RH pollution periods. Aqueous-phase production of $SO_4^{2-}$ become important
during those periods. Consistently, Fig. 7 shows that the sulfate oxidation ratio (SOR =
$[SO_4^{2-}]/([SO_4^{2-}] + [SO_2])$) increased exponentially with the increase of ALWC at RH >50%.
A strong correlation of the mass concentrations between OOA and $NO_3^-$ was observed with
$R^2$ of 0.84 (Fig. 8a), possibly explained by the dominant contribution of photochemical
production for both OOA and $NO_3^-$. When considering the RH effect (color coded in Fig.
8a), it is found that the data are scattered around the regression line with uniform slope
when RH <70% but concentrated in a small area above the regression line when RH >70%,
suggesting that the OOA formation at RH >70% is probably promoted by aerosol water.
This is further supported by the linear increase of OOA with increasing $SO_4^{2-}$ when RH
>70%, while the relationship between OOA and $SO_4^{2-}$ was very scattered when RH <70%
(Fig. 8b).
**4 Conclusion**
We conducted online measurements of $PM_1$ in urban Beijing from 29 December 2014 to
27 February 2015. The average mass concentration of $PM_1$ was 73.8 μg m$^{-3}$ and OA was the
most important component of $PM_1$ (52%), followed by nitrate (14%) and sulfate (10%).
Source apportionment of OA resolved five factors including HOA, COA, BBOA, CCOA, and
OOA, in which CCOA (32%) and OOA (32%) were the most important sources to OA. The
mass proportion of CCOA in OA showed a significant increase from clean period (25%) to
pollution periods (31-35%), highlighting the important role of coal burning in haze
formation in wintertime Beijing. The meteorological conditions (WD, WS, and RH) have a
significant impact on the chemical composition and evolution of $PM_1$ species. Nitrate had
a higher contribution during low-RH pollution days, implying the photochemical



oxidation process of nitrate formation. In contrast, the mass fraction of sulfate to $PM_1$ was
much higher during high-RH pollution episodes compared to those during low-RH
pollution episodes. The data also showed the exponential increase of sulfate oxidation
ratio (SOR) with ALWC at high RH conditions. Both are consistent with the impacts of
aqueous-phase reactions on the formation of sulfate. As for the OOA formation, the strong
correlation between OOA and $NO_3^-$ may be explained by the dominant role of
photochemical production on both species; aqueous-phase processes may add an
additional contribution to OOA formation under high RH condition, as indicated by the
linear increase of OOA with increasing $SO_4^{2-}$ when RH >70%.
*Data availability.* Raw data used in this study are archived at the Institute of Earth
Environment, Chinese Academy of Sciences, and are available on request by contacting
the corresponding author.
*Supplement.* The Supplement related to this article is available online at
*Competing interests.* The authors declare that they have no conflict of interest.
*Author contributions.* RJH designed the study. Data analysis and interpretation were made
by YH, JD, and RJH. RJH, JD, and YH prepared the manuscript with contributions from all
authors.
*Acknowledgments.* This work was supported by the National Natural Science Foundation
of China (NSFC) under Grant No. 41925015, 91644219, 41877408 and 41675120, the
National Key Research and Development Program of China (No. 2017YFC0212701), the
Chinese Academy of Sciences (no. ZDBS-LY-DQC001), and the Cross Innovative Team fund
from the State Key Laboratory of Loess and Quaternary Geology (No. SKLLQGTD1801),
and the Irish Environmental Protection Agency and Science Foundation Ireland project of
OM-MaREI.





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





**Table1** Summary of the $PM_1$ composition, OA sources and meteorological conditions
during different pollution periods.

| Species | Clean | High-RH pollution | Low-RH pollution |
|---|---|---|---|
| $PM_1$ ($\mu g\ m^{-3}$) | 19.5 | 123.2 | 125.4 |
| Org ($\mu g\ m^{-3}$) | 10.9 (56%) | 56.7 (46%) | 67.7 (54%) |
| $SO_4^{2-}$ ($\mu g\ m^{-3}$) | 2.0 (10%) | 20.9 (17%) | 8.8 (7%) |
| $NO_3^-$ ($\mu g\ m^{-3}$) | 2.2 (11%) | 17.2 (14%) | 18.8 (15%) |
| $NH_4^+$ ($\mu g\ m^{-3}$) | 1.8 (9%) | 12.3 (10%) | 11.3 (9%) |
| $Cl^-$ ($\mu g\ m^{-3}$) | 1 (5%) | 7.4 (6%) | 8.8 (7%) |
| BC ($\mu g\ m^{-3}$) | 1.7 (9%) | 8.6 (7%) | 10.0 (8%) |
| HOA ($\mu g\ m^{-3}$) | 0.8 (8%) | 9.1 (16%) | 8.8 (13%) |
| COA ($\mu g\ m^{-3}$) | 2 (20%) | 6.8(12%) | 8.8 (13%) |
| BBOA ($\mu g\ m^{-3}$) | 1 (10%) | 3.4 (6%) | 8.1 (12%) |
| CCOA ($\mu g\ m^{-3}$) | 2.8 (25%) | 17.6 (31%) | 23.7 (35%) |
| OOA ($\mu g\ m^{-3}$) | 4.1 (37%) | 19.8 (35%) | 18.3 (27%) |
| $O_x$ (ppb) | 39.2 | 47.8 | 59.8 |
| $NO_2$ (ppb) | 16.7 | 64.3 | 103.0 |
| RH (%) | 25 | 60 | 31 |
| WS ($m\ s^{-1}$) | 2.5 | 1 | 0.9 |
| Vis (Km) | 15.7 | 6.5 | 6.7 |










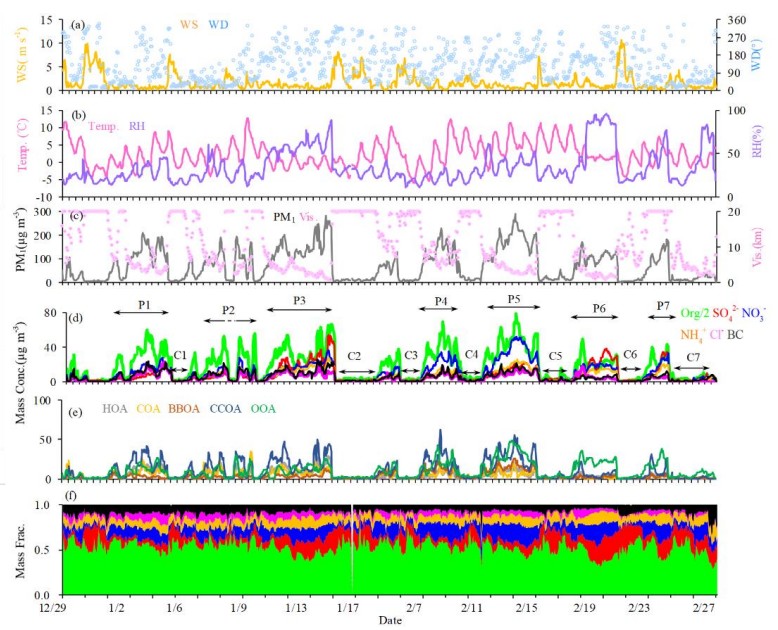


**Figure 1.** Time series of (a) wind speed (WS) and wind direction (WD), (b) Temperature (Temp) and relative humidity (RH), (c) visibility and PM$_1$, (d) NR-PM$_1$ species (i.e., OA, SO$_4^{2-}$, NO$_3^-$, NH$_4^+$, Cl$^-$ and BC; note that OA is halved clarity), (e) OA factors (i.e., HOA, COA, BBOA, CCOA and OOA), and (f) relative contribution of PM$_1$ species.














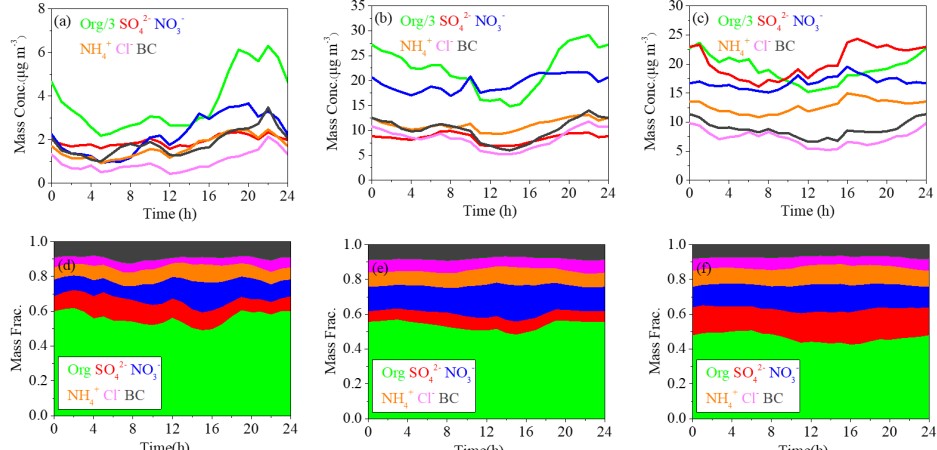


**Figure 2.** The diurnal variations of mass concentrations and relative contributions of $PM_1$
components during clean days (a, d), low-RH pollution days (b, e) and high-RH pollution
days (c, f).


















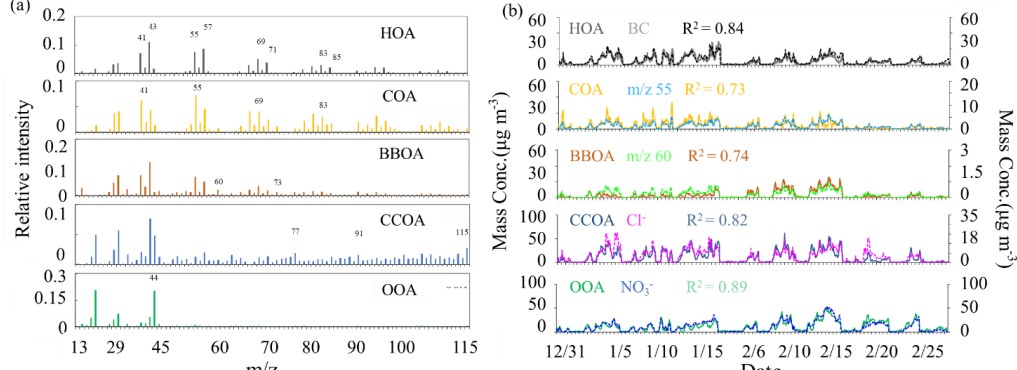


**Figure 3.** The mass spectra(a) and time series(b) of OA factors (HOA, COA, BBOA, CCOA, and OOA).

814                    .

















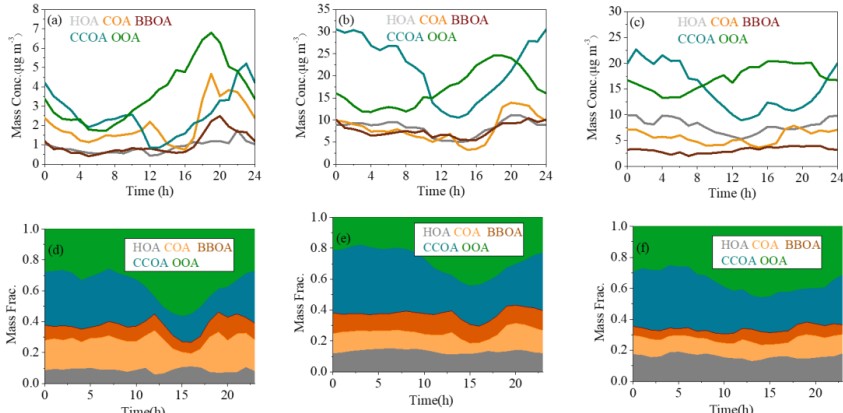


**Figure 4.** The diurnal variations of mass concentrations and relative contributions of OA
factors during clean days (a, d), low-RH pollution days (b, e) and high-RH pollution days
(c, f).






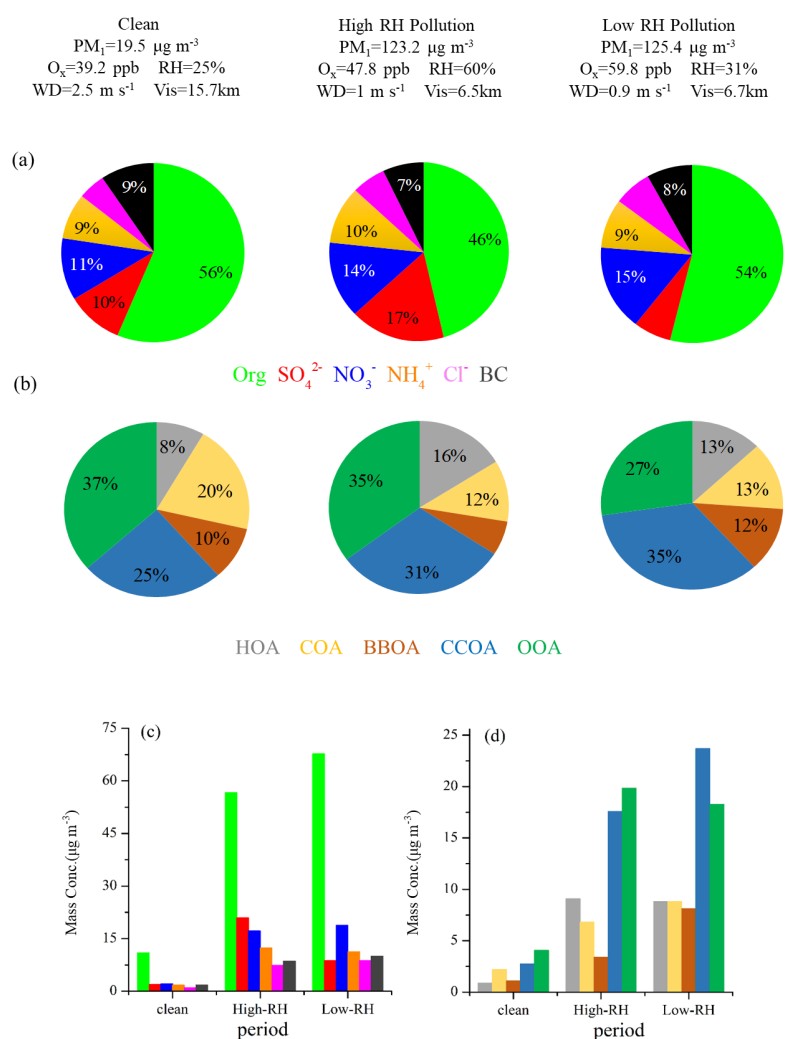


**Figure 5.** PM$_1$ chemical composition (a) and OA source composition (b) pie chart as well
as the mass concentrations of PM$_1$ species(c) and OA sources(d) during clean, High-RH
pollution and Low-RH pollution periods.






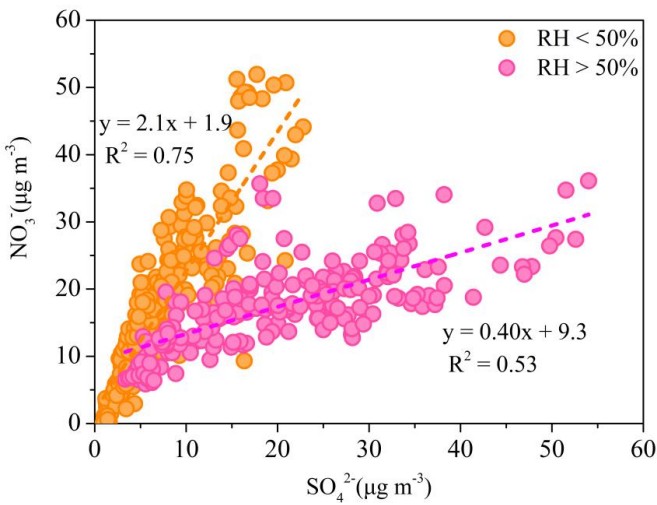


**Figure 6.** The relationship between $SO_4^{2-}$ and $NO_3^-$ during low-RH (RH <50%) and high-RH (RH >50%) pollution episodes.











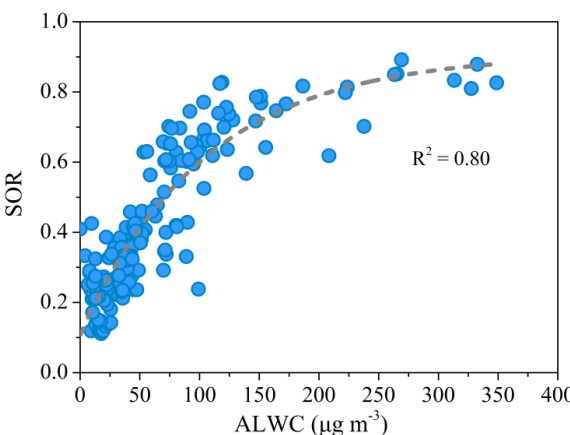


**Figure 7.** The relationship between the sulfate oxidation ratio (SOR = $[SO_4^{2-}]/([SO_4^{2-}]$ + $[SO_2])$) and ALWC at high RH condition (RH >50%).














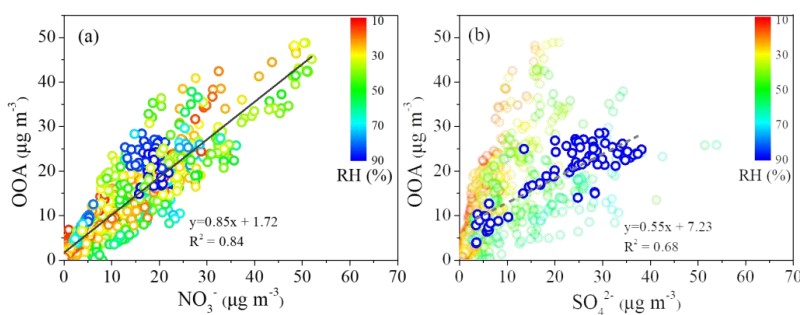


**Figure 8.** Scatter plot between the mass concentration of OOA and $NO_3^-$ (colored by RH)
(a), and scatter plot between the mass concentration of OOA and $SO_4^{2-}$ (colored by RH)
(b).