# Peer review of "Contrasting sources and processes of particulate species in haze days with low and"

_Atmospheric Chemistry and Physics, 2020_

## Referee Comment (RC1) · Anonymous Referee #1 · 31 Mar 2020

In this study, Huang et al. used an aerosol chemical species monitor (ACSM) and an aethalometer to characterize the organic aerosol (OA) in wintertime Beijing. Positive matrix factorization (PMF) was applied to resolve the sources and processes for OA. The effect of RH on the mass concentration, the mass fraction, and the growth rate of various components in PM1 was analyzed. OA was found to dominate the components under both high-RH and low-RH pollution periods. But the change of sulfate and nitrate showed opposite RH-dependence. The results demonstrated the importance of photochemical oxidation and aqueous-phase processes on the formation of secondary aerosol during haze episodes. It could be helpful for understanding the haze formation in wintertime Beijing. Overall, the results are well presented. I recommend

the manuscript be considered for publication after following comments being fully addressed. 1. First of all, the motivation of research should be explained more clearly in the introduction section. 2. The criterions to distinguish clean day and polluted days, as well as low RH and high RH seem a little bit arbitrary. The selection of the concentration threshold of PM1 for discriminating the clean and pollution period, as well as RH, should be explained. 3. It was noted that in Figure 4 where OOA reached a peak value at about 20:00 LT. The authors may give an interpretation about this phenomenon. Also, what is the reason that OOA increased more quickly under low RH condition than under high RH conditions? 4. Lines 328-332: Why the concentration of Ox (=O3+NO2) was lower than that of NO2? Please check carefully. Meanwhile, the mass fraction of some species (e.g., Cl-) was missing in Fig. 5. 5. Lines 335-337: I don't think there was obvious difference in the mass fraction of nitrate in two types of pollution periods (14 % vs. 15 %). And references need to be cited in Line 337. In addition, this sentence is too hard to read. Rewrite it! 6. Lines 391-395: Was P3 assigned to high-RH or low-RH pollution during the analyses of sulfate and OA? 7. Line 423. The measurement of SO2 was not described in the Section 2.1 Site description and instrumentation. 8. Conclusion. I think a brief description of atmospheric implications should be included in this section.

---

## Referee Comment (RC2) · Anonymous Referee #2 · 31 Mar 2020

This manuscript presents the comparisons of PM1 species and organic aerosol (OA) sources/processes during winter in Beijing for clean and pollution periods, with a particular focus on the effect of relative humidity (RH) on secondary aerosol formation. The comparisons were made through mass concentration, mass fraction, and growth rate. It is found that OA dominated the PM1 mass under both low-RH and high-RH pollution conditions. However, sulfate was found to increase during high-RH pollution periods and nitrate increased during low-RH pollution periods. Oxygenated OA (OOA) showed higher growth rate during low-RH pollution period than during high-RH pollution period. These results provide insights into the relative importance of photochemical oxidation vs. aqueous-phase processes for secondary aerosol formation under different meteorological conditions. It is a useful addition to the literature for understanding the haze formation in Beijing. The manuscript is well written, and results are discussed logically. I recommend publication in ACP after a few minor points are addressed.

(1) In section 2.2.3, organics was not considered in the ALWC calculation using the ISORROPIA-II model. Please provide an explanation.

(2) Page 6, line 203-205, the authors divided the pollution period into low-RH pollution days (RH <50%) and high-RH pollution days (RH >50%). What is the criterion for this definition of low- and high-RH?

(3) Page 8, line 303, change "OOA is correlated well with nitrate (R2=0.89). and the diurnal cycle . . . . . ." to "OOA is correlated well with nitrate (R2=0.89), and the diurnal cycle . . . . . .".

(4) Page 9, line 330-334. ". . . . .We observed a much larger contribution from nitrate during low-RH pollution periods than during high-RH pollution periods. . . . . ." This is not well supported as both the mass concentration and fraction of nitrate are similar. Please check it carefully.

(5) In Table 1 and throughout the manuscript, the authors should pay attention to the significant digits which denotes precision of measurements.

---

## Referee Comment (RC3) · Anonymous Referee #3 · 1 Apr 2020

This paper focuses on discussing the secondary aerosol formation processes under different RH conditions mainly. It is valuable to understand the aerosol chemistry in Beijing, but I have a few serious concerns on current results and interpretation. See below:

(1) One concern is that the dataset used is relatively old (five years ago). It is therefore not very up-to-date to reflect the real processes in current atmosphere given the concentrations, compositions of PM1 as well as the precursors might have changed greatly in Beijing. The authors have to comment more on the implications of findings here.

[Figure]

(2) More details regarding the PMF analyses should be provided. It is not clear why 5-factor solution is optimal, considering that you use ACSM data which had very limited chemical resolution to identify tracer ions. And you used ME-2 technique, which on one hand is better to extract the real factors presenting in your data, but on the other hand, you may presume and artificially identify a factor that might not be real. Justification of the PMF results is essential and why and how the initial profiles of different factors were used are not clear. I feel that current information provided here is not enough.

(3) Calculation of ALWC by using ISORROPIA-II model had uncertainties as it only consider water uptake by inorganic species but organics is dominant your PM1, please comment on this, and discuss the influences on your results.

(4) A very serious concern is the large uncertainty of your interpretation. In order to make strong argument regarding the different chemical processes under different RH conditions. You have to eliminate the influences of other meteorological conditions (PBL, wind directions, speeds, and different air masses) on the concentrations, compositions and growth rates you investigated here. Otherwise, you cannot claim that the observed changes were solely due to chemistry. This reviewer see very little discussions regarding this point, and this make the results highly untrustworthy. For example, the calculation of growth rates, such rates is largely not due to chemistry but likely PBL variations, etc. In understand that the authors argue that during pollution period, there was low wind and mainly south/southeasterly wind mainly; this is too general and does not help resolve what I mention here.

(5) L199-L201: Table 1 does not provide wind directions as you said.

(6) Indeed, similar discussion had been published in a few references cited here, and it seems to be a bit superficial here, especially section 3.4. The authors need to add more discussions, and point out clearly what are the unique and novel findings here from other studies.

(7) Why you chose 50% RH as a cutting point for low- and high-RH conditions? How

about 60%, and how does this choice possibly influence your findings?

---

## Author Comment (AC1) · 8 Jun 2020

The authors thank the editor and referees to review our manuscript and particularly for the valuable comments and suggestions that are very helpful in improving the manuscript. We provide below point-by-point responses to those comments. We also have made most of the changes suggested by the referees in the revised manuscript.

Please also note the supplement to this comment: https://www.atmos-chem-phys-discuss.net/acp-2020-158/acp-2020-158-AC1-supplement.pdf

**Supplement:**

The authors thank the editor and referees to review our manuscript and particularly for the valuable comments and suggestions that are very helpful in improving the manuscript. We provide below point-by-point responses to those comments. We also have made most of the changes suggested by the referees in the revised manuscript.

**Referee #1**

In this study, Huang et al. used an aerosol chemical species monitor (ACSM) and an aethalometer to characterize the organic aerosol (OA) in wintertime Beijing. Positive matrix factorization (PMF) was applied to resolve the sources and processes for OA. The effect of RH on the mass concentration, the mass fraction, and the growth rate of various components in PM1 was analyzed. OA was found to dominate the components under both high-RH and low-RH pollution periods. But the change of sulfate and nitrate showed opposite RH-dependence. The results demonstrated the importance of photochemical oxidation and aqueous-phase processes on the formation of secondary aerosol during haze episodes. It could be helpful for understanding the haze formation in wintertime Beijing. Overall, the results are well presented. I recommend the manuscript be considered for publication after following comments being fully addressed.

(1) First of all, the motivation of research should be explained more clearly in the introduction section.
**Response:** We thank the referee's suggestion. In the revised manuscript lines 103-107, we have now added the following description: "Despite the observations of large production of secondary aerosol during haze events, the formation mechanisms are not yet well understood. Specifically, more studies are needed to elucidate the relative importance of photochemical oxidation versus aqueous-phase processes on the formation of secondary aerosol during wintertime haze episodes of different meteorological conditions".

(2) The criterions to distinguish clean day and polluted days, as well as low RH and high RH seem a little bit arbitrary. The selection of the concentration threshold of PM1 for discriminating the clean and pollution period, as well as RH, should be explained.
**Response:** As shown in Figure 1, the clean and pollution episodes occurred alternately during the measurement period, and the $PM_1$ concentration was usually lower than 20 µg m$^{-3}$ during clean episodes and higher than 100 µg m$^{-3}$ during polluted episodes. Thus, we divided the measurements into clean period and pollution period using the criteria of 20 µg m$^{-3}$ and 100 µg m$^{-3}$, respectively. Furthermore, during the pollution period, RH varied from 15% to 95% with an average value of 46% ($\approx$ 50%) and a median value of 43%, thus we used 50% as the criterion to further divide the pollution period into low-RH pollution days (RH <50%) and high-RH pollution days (RH >50%).

In lines 213-215 in the revised manuscript, we have now added "The clean and pollution episodes occurred alternately during the measurement period, and the $PM_1$ concentration was usually lower than 20 µg m$^{-3}$ during clean episodes and higher than 100 µg m$^{-3}$ during pollution episodes. As such…".

And in lines 220-221, we have added "During the polluted period, RH varied from 15% to 95% with an average value of 46% and a median value of 43%. To investigate…".

(3) It was noted that in Figure 4 where OOA reached a peak value at about 20:00 LT. The authors may give an interpretation about this phenomenon. Also, what is the reason that OOA increased more quickly under low RH condition than under high RH conditions?

**Response:** In the section 3.2, we discuss that the diurnal cycle of OOA shows an increase from about 6:00 to 20:00 LT, indicating the contribution from photochemical production. The peak value at about 20:00 LT may be due to the accumulation of OOA formed from photochemical production. Meanwhile, as discussed in section 3.4, photochemical production contributed dominantly to OOA. The $O_x$ concentration during low-RH pollution days (59.8 ppb) was higher than that during high-RH pollution days (47.8 ppb) and clean days (39.2 ppb). With the higher $O_x$ concentration (as a surrogate of oxidant level) under low-RH conditions, the daytime formation of OOA was more efficient and the growth rate was higher during those low-RH pollution days than those during high-RH pollution days and clean days.

In line 328 of the revised manuscript, we added "…photochemical production and accumulation of OOA".

(4) Lines 328-332: Why the concentration of Ox (=O3+NO2) was lower than that of NO2? Please check carefully. Meanwhile, the mass fraction of some species (e.g., Cl-) was missing in Fig. 5.

**Response:** Apologies for the typos. We double check our data, and it now reads: "$NO_2$ increased accordingly from 16.7 ppb during clean periods to 42.2 ppb during high-RH pollution periods and to 55.4 ppb during low-RH pollution periods". We have changed the $NO_2$ values in Table1 accordingly. The missing mass fractions for some species are also added in Figure 5.

(5) Lines 335-337: I don't think there was obvious difference in the mass fraction of nitrate in two types of pollution periods (14 % vs. 15 %). And references need to be cited in Line 337. In addition, this sentence is too hard to read. Rewrite it!

**Response:** We thank the referee for pointing this out. We have made the change and it now reads "We observed similar contributions from nitrate during low-RH pollution periods and high-RH pollution periods, while a much larger contribution from sulfate during high-RH pollution periods than during low-RH pollution periods because of enhanced formation from aqueous-phase processes".

Meanwhile, references were added in line 363, and it now reads "…indicating the importance of residential coal combustion emissions during haze pollution in wintertime Beijing (Elser et al., 2016; Li et al., 2017).

(6) Lines 391-395: Was P3 assigned to high-RH or low-RH pollution during the analyses of sulfate and OA?

**Response:** P3 was assigned to high-RH pollution episode during the analyses of sulfate

and OOA. We made change in lines 415-416 and it now reads "OOA also showed a much higher contribution to OA during high-RH pollution events (62% for P6 and 50% for P7) than during low-RH pollution events (P1, P2, P4 and P5, 20-31%)".

(7) Line 423. The measurement of SO2 was not described in the Section 2.1 Site description and instrumentation.
**Response:** Thanks for the referee's reminder. In the revised manuscript lines 136-138, we have now added "The gaseous species including $O_3$, $NO_x$, and $SO_2$ were measured by a Thermo Scientific Model 49i ozone analyzer, a Thermo Scientific Model 42i $NO–NO_2–NO_x$ analyzer, and an Ecotech EC 9850 sulfur dioxide analyzer, respectively".

(8) Conclusion. I think a brief description of atmospheric implications should be included in this section.

**Response:** In the revised manuscript (lines 478-481), we have now added the following discussion "These results provide insights into the relative importance of photochemical oxidation and aqueous-phase processes for secondary aerosol formation during haze pollution, demonstrating the significance of meteorological conditions in the formation of secondary aerosol."

**Referee #2**

This manuscript presents the comparisons of PM1 species and organic aerosol (OA) sources/processes during winter in Beijing for clean and pollution periods, with a particular focus on the effect of relative humidity (RH) on secondary aerosol formation. The comparisons were made through mass concentration, mass fraction, and growth rate. It is found that OA dominated the PM1 mass under both low-RH and high-RH pollution conditions. However, sulfate was found to increase during high-RH pollution periods and nitrate increased during low-RH pollution periods. Oxygenated OA (OOA) showed higher growth rate during low-RH pollution period than during high-RH pollution period. These results provide insights into the relative importance of photochemical oxidation vs. aqueous-phase processes for secondary aerosol formation under different meteorological conditions. It is a useful addition to the literature for understanding the haze formation in Beijing. The manuscript is well written, and results are discussed logically. I recommend publication in ACP after a few minor points are addressed.

(1) In section 2.2.3, organics was not considered in the ALWC calculation using the ISORROPIA-II model. Please provide an explanation.
**Response:** We thank the reviewer for pointing this out. We calculated the ALWC using the ISORROPIA-II model, which simulates the thermodynamic equilibrium of the $NH_4^+–SO_4^{2-}–NO_3^-–Cl^-–H_2O$ system and does not consider the organics contribution. To evaluate this uncertainty, we further calculate the contribution of organics to ALWC following the

approach in Guo et al (2015) and Cheng et al (2016):

$$W_{org} = \frac{OM}{\rho_{org}} \cdot \rho_w \cdot \frac{\kappa_{org}}{(100\%/RH - 1)}$$

where OM is the mass concentration of organics, $\rho_w$ is the density of water and $\rho_{org}$ is the density of organics ($\rho_{org}$ =1.4 × 10$^3$ kg m$^{-3}$, Cerully et al., 2014). $\kappa_{org}$ is the hygroscopicity parameter of organic aerosol composition. We adopted a $\kappa_{org}$ value of 0.06 based on previous cloud condensation nuclei measurements in Beijing (Gunthe et al., 2011).

The calculated results showed that the average contribution of organics to the total ALWC was 18%, suggesting that inorganic species were the dominant hygroscopic species and organics had a minor contribution to ALWC.

In the revised manuscript, we added the following description in section 2.3.3: "Meanwhile, the contribution of organics to ALWC (ALWC$_O$) was also calculated using the following equation (Guo et al., 2015; Cheng et al., 2016):

$$W_{org} = \frac{OM}{\rho_{org}} \cdot \rho_w \cdot \frac{\kappa_{org}}{(100\%/RH - 1)}$$

where OM is the mass concentration of organics, $\rho_w$ is the density of water and $\rho_{org}$ is the density of organics ($\rho_{org}$ =1.4 × 10$^3$ kg m$^{-3}$, Cerully et al., 2014). $\kappa_{org}$ is the hygroscopicity parameter of organic aerosol composition. We adopted a $\kappa_{org}$ value of 0.06 based on previous cloud condensation nuclei measurements in Beijing (Gunthe et al., 2011)."
Figure 7 was also updated accordingly.

(2) Page 6, line 203-205, the authors divided the pollution period into low-RH pollution days (RH <50%) and high-RH pollution days (RH >50%). What is the criterion for this definition of low- and high-RH?
**Response:** As discussed in response to referee #1, during the polluted period, RH varied from 15% to 95% with an average value of 46% ($\approx$50%) and a median value of 43%. Thus, we used 50% as the criterion to further divide the pollution period into low-RH pollution days (RH <50%) and high-RH pollution days (RH >50%). If 60% is used as a cutting point, the data points (78) in RH >60% are much less than those in RH<60% (282), which may be not proper for statistical comparison.

In the revised manuscript lines 220-221, we have added "During the polluted period, RH varied from 15% to 95% with an average value of 46% and a median value of 43%. To investigated…".

(3) Page 8, line 303, change "OOA is correlated well with nitrate (R2=0.89). and the diurnal cycle : : :: : :" to "OOA is correlated well with nitrate (R2=0.89), and the diurnal cycle : : :: : :".
**Response:** Change made.

(4) Page 9, line 330-334. ": : :: : :We observed a much larger contribution from nitrate

during low-RH pollution periods than during high-RH pollution periods: : :: : :" This is not well supported as both the mass concentration and fraction of nitrate are similar. Please check it carefully.

**Response:** We thank the referee for pointing this out. In the revised manuscript lines 353-358, it now reads "We observed similar contributions from nitrate during low-RH pollution periods and high-RH pollution periods, while a much larger contribution from sulfate during high-RH pollution periods than during low-RH pollution periods because of enhanced formation from aqueous-phase processes".

(5) In Table 1 and throughout the manuscript, the authors should pay attention to the significant digits which denotes precision of measurements.

**Response:** Thanks. In the revised manuscript, we have now unified the significant digits throughout the manuscript and Table 1.

**Referee #3**

This paper focuses on discussing the secondary aerosol formation processes under different RH conditions mainly. It is valuable to understand the aerosol chemistry in Beijing, but I have a few serious concerns on current results and interpretation. See below:

(1) One concern is that the dataset used is relatively old (five years ago). It is therefore not very up-to-date to reflect the real processes in current atmosphere given the concentrations, compositions of PM1 as well as the precursors might have changed greatly in Beijing. The authors have to comment more on the implications of findings here.

**Response:** We agree with the referee that the concentrations, compositions of $PM_1$ as well as the precursors might have some changes in recent years. Our measurements were conducted after the implementation of legislative 'Air Pollution Prevention and Control Action Plan' in 2013. Therefore, our results show some similar variations on the $PM_1$ composition when compared to more recent studies, such as the increase of nitrate and decrease of sulfate. Meanwhile, our study focused on the formation mechanisms of secondary aerosol during different meteorological conditions in haze pollution, especially the significant effects of RH, which is still not well elucidated. The results demonstrated the importance of photochemical oxidation and aqueous-phase processes on the formation of secondary aerosol during haze episodes. It could be helpful for understanding the haze formation in wintertime Beijing. Therefore, our study still provides valuable information to the scientific community to improve our understanding of fine PM pollution.

In the revised manuscript, we have now added the following discussion in conclusion: "These results provide insights into the relative importance of photochemical oxidation and aqueous-phase processes for secondary aerosol formation during haze pollution, demonstrating the significance of meteorological conditions in determining the formation of secondary aerosol".

(2) More details regarding the PMF analyses should be provided. It is not clear why 5-factor solution is optimal, considering that you use ACSM data which had very limited chemical resolution to identify tracer ions. And you used ME-2 technique, which on one hand is better to extract the real factors presenting in your data, but on the other hand, you may presume and artificially identify a factor that might not be real. Justification of the PMF results is essential and why and how the initial profiles of different factors were used are not clear. I feel that current information provided here is not enough.

**Response:** Thanks for the referee's suggestion. We first performed free PMF (unconstrained PMF) runs to identify the main factors. The number of factors was determined by examining three to seven factors, and the factors were identified based on their mass spectral profiles and correlation with external tracers. The five-factor solution (i.e., HOA, COA, CCOA, BBOA, and OOA) was selected because it best explains our data. For the solutions with less than 5 factors, HOA appeared to be mixed with COA while CCOA mixed with BBOA (Fig. S1). However, when the number of factors was increased to 6, the OOA factor split into two OOA factors of similar time series (Fig. S2), suggesting that further separation of the factors does not improve the interpretation of the data. After determining the number of factors (five) using un-constrained PMF runs, we then performed ME-2 runs, which explore the variabilities of certain (mainly primary) factors (i.e., HOA, COA, and BBOA) by constraining their profiles with *a* value approach. In a word, the un-constraint PMF helps making sure that the five factors were real (without "forced deconvolution" as in constrained ME-2), while the constrained ME-2 helps refining the contributions from those five factors. Please see Fig. S1-3 and the corresponding discussion in Section 3.2 for further details of the PMF/ME-2.

(3) Calculation of ALWC by using ISORROPIA-II model had uncertainties as it only consider water uptake by inorganic species but organics is dominant your PM1, please comment on this, and discuss the influences on your results.

**Response:** We thank the reviewer for pointing this out. We calculated the ALWC using the ISORROPIA-II model, which simulates the thermodynamic equilibrium of the $NH_4^+$–$SO_4^{2-}$–$NO_3^-$–$Cl^-$–$H_2O$ system and does not consider the organics contribution. To evaluate this uncertainty, we further calculate the contribution of organics to ALWC following the approach in Guo et al (2015) and Cheng et al (2016):

$$W_{org} = \frac{OM}{\rho_{org}} \cdot \rho_w \cdot \frac{\kappa_{org}}{(100\%/RH - 1)}$$

where OM is the mass concentration of organics, $\rho_w$ is the density of water and $\rho_{org}$ is the density of organics ($\rho_{org}$ =1.4 × 10$^3$ kg m$^{-3}$, Cerully et al., 2014). $\kappa_{org}$ is the hygroscopicity parameter of organic aerosol composition. We adopted a $\kappa_{org}$ value of 0.06 based on previous cloud condensation nuclei measurements in Beijing (Gunthe et al., 2011).

The calculated results showed the average contribution of organics to the total ALWC was 18%, suggesting that inorganic species were the dominant hygroscopic species and organics had minor contribution to ALWC.

In the revised manuscript, we have added following description in section 2.3.3:
"Meanwhile, the contribution of organics to ALWC(ALWC$_0$) was also calculated using the following equation (Guo et al., 2015; Cheng et al., 2016):

$$W_{org} = \frac{OM}{\rho_{org}} \cdot \rho_w \cdot \frac{\kappa_{org}}{(100\%/RH - 1)}$$

where OM is the mass concentration of organics, $\rho_w$ is the density of water and $\rho_{org}$ is the density of organics ($\rho_{org}$ =1.4 × 10$^3$ kg m$^{-3}$, Cerully et al., 2014). $\kappa_{org}$ is the hygroscopicity parameter of organic aerosol compositions. We adopted a $\kappa_{org}$ value of 0.06 based on previous cloud condensation nuclei measurements in Beijing (Gunthe et al., 2011)."
Also Figure 7 was updated accordingly.

(4) A very serious concern is the large uncertainty of your interpretation. In order to make strong argument regarding the different chemical processes under different RH conditions. You have to eliminate the influences of other meteorological conditions (PBL, wind directions, speeds, and different air masses) on the concentrations, compositions and growth rates you investigated here. Otherwise, you cannot claim that the observed changes were solely due to chemistry. This reviewer see very little discussions regarding this point, and this make the results highly untrustworthy. For example, the calculation of growth rates, such rates is largely not due to chemistry but likely PBL variations, etc. In understand that the authors argue that during pollution period, there was low wind and mainly south/southeasterly wind mainly; this is too general and does not help resolve what I mention here.

**Response:** We agree with the referee that factors, such as PBL height and winds, might complicate the interpretation of chemical processes under different RH conditions. For example, the growth rates of different species might be affected by those factors. Here in this study, however, we did not intend to emphasize on the absolute values of growth rates, which make less sense given the factors mentioned above. Instead, we mostly tried to compare the growth rates of different species (e.g., sulfate and nitrate in lines 243-244) under the same periods, which should share the effects by those factors. When we did compare the growth rates of the same species under different time periods (low-RH and high-RH pollution conditions), as also pointed out by the reviewer, we assumed that the influences from those factors were more or less similar for these two types of pollution conditions. To provide a note of caution, the caveat of this kind of comparison is now added in lines 250-255 in the revised manuscript: "Note that the comparison of growth rates was done under the assumption that chemical processes were the main reason for mass growth, which might not be the case if other factors such as planetary boundary layer height variations dominate. Yet comparison of growth rates of different species in the same time period would not be affected by these factors because those species should share the same effects".

(5) L199-L201: Table 1 does not provide wind directions as you said.
**Response:** Apologies for the typo. Table 1 has been changed to Fig. 1.

(6) Indeed, similar discussion had been published in a few references cited here, and it seems to be a bit superficial here, especially section 3.4. The authors need to add more discussions, and point out clearly what are the unique and novel findings here from other studies.

**Response:** We agree with the referee that some similar discussion had been published in a few references cited here. However, the causes of fine PM pollution in urban Beijing are still not fully understood, most likely due to the campaign-to-campaign difference in meteorological conditions, emissions, and atmospheric processes. In addition, previous studies usually focused on sources variations or only SIA or SOA formations (Sun et al., 2015; Hu et al., 2016; Hu et al., 2017; Xu et al., 2017). In our study, we focused on the RH effects during haze pollution. The effect of RH on the mass concentration, the mass fraction, and the growth rate of various components in $PM_1$ was all analyzed. OA was found to dominate the components under both high-RH and low-RH pollution periods. But the change of sulfate and nitrate showed opposite RH-dependence. The results demonstrated the importance of photochemical oxidation and aqueous-phase processes on the formation of secondary aerosol during haze episodes. It could be helpful for understanding the haze formation in wintertime Beijing. Therefore, our study still provides valuable information to the scientific community to improve our understanding of fine PM pollution.

In the revised manuscript, we have now added the following discussion in section 3.4 in lines 436-438: "Meanwhile, the high ratio between $NO_3^-$ and $SO_4^{2-}$ suggest the nitrate production is more efficient than that of sulfate during low-RH pollution period", and in lines 447-452: "Meanwhile, the $O_x$ concentration during low-RH pollution days (59.8 ppb) was higher than that during high-RH pollution days (47.8 ppb) and clean days (39.2 ppb). With the higher $O_x$ concentration (as a surrogate of oxidant level) under low-RH conditions, the daytime formation of OOA was more efficient and the growth rate was higher during those low-RH pollution days than those during high-RH pollution days and clean days".

And added the following discussion in the conclusion in lines 478-481: "These results provide insights into the relative importance of photochemical oxidation and aqueous-phase processes for secondary aerosol formation during haze pollution, demonstrating the significance of meteorological conditions in determining the formation of secondary aerosol".

(7) Why you chose 50% RH as a cutting point for low- and high-RH conditions? How about 60%, and how does this choice possibly influence your findings?

**Response:** During the pollution period, RH varied from 15% to 95% with an average value of 46% and a median value of 43%. Thus, we used 50% as the criterion to further divide the pollution period into low-RH pollution days (RH <50%) and high-RH pollution days (RH >50%). If 60% is used as a cutting point, the data points (78) in RH >60% are much less than those in RH<60% (282), which may be not proper for statistical comparison. Meanwhile, this choice has minor influence on our findings. For example, when using 60% RH as the cutting point, Figure R1 (left) and Figure R2 (right) are very similar to those

two figures (Fig. 6 and Fig. 7) in the manuscript. The better correlation between SOR and ALWC in high RH condition and the conclusion of "Aqueous-phase production of $SO_4^{2-}$ become important during high RH periods" are still valid. Also, as shown in Figure R1 (left) below, when using 60% RH, some data in 50%~60% shows similar trend with that of RH>60%, further confirming that 50% RH is a better choice as the cutting point for low- and high-RH pollution conditions.

[Figure]

Fig. R1. The relationship between $SO_4^{2-}$ and $NO_3^-$ during low-RH (RH <60%) and high-RH (RH >60%) pollution episodes (left).
Fig. R2. The relationship between the sulfate oxidation ratio (SOR = $[SO_4^{2-}]/([SO_4^{2-}] + [SO_2])$) and ALWC at high RH pollution condition (RH >60%) (right).

In the revised manuscript lines 220-221, we added "During the polluted period, RH varied from 15% to 95% with an average value of 46% and a median value of 43%. To investigate…".

Reference:
Cerully, K. M., Bougiatioti, A., Hite Jr., J. R., Guo, H., Xu, L., Ng, N. L., Weber, R., and Nenes, A.: On the link between hygroscopicity, volatility, and oxidation state of ambient and water-soluble aerosols in the southeastern United States, Atmos. Chem. Phys., 15, 8679–8694, https://doi.org/10.5194/acp-15-8679-2015, 2015.
Cheng, Y. F., Zheng, G. J., Wei, C., Mu, Q., Zheng, B., Wang, Z. B., Gao, M., Zhang, Q., He, K. B., Carmichael, G., Pöschl, U., and Su, H.: Reactive nitrogen chemistry in aerosol water as a source of sulfate during haze events in China, Sci. Adv., 2, e1601530, https://doi.org/10.1126/sciadv.1601530, 2016.
Elser, M., Huang, R. J., Wolf, R., Slowik, J. G., Wang, Q., Canonaco, F., Li, G., Bozzetti, C., Daellenbach, K. R., Huang, Y., Zhang, R., Li, Z., Cao, J., Baltensperger, U., El-Haddad, I., and Prévôt, A. S. H.: New insights into PM2.5 chemical composition and sources in two major cities in China during extreme haze events using aerosol mass spectrometry, Atmos. Chem. Phys., 16, 3207–3225, https://doi.org/10.5194/acp-16-3207-2016, 2016.
Gunthe, S. S., Rose, D., Su, H., Garland, R. M., Achtert, P., Nowak, A., Wiedensohler, A., Kuwata, M., Takegawa, N., Kondo, Y., Hu, M., Shao, M., Zhu, T., Andreae, M. O., and Poschl, U.:Cloud condensation nuclei (CCN) from fresh and aged air pollution in the megacity region of

Beijing, Atmos. Chem. Phys., 11, 11023–11039, 2011.

Guo, H., Xu, L., Bougiatioti, A., Cerully, K.M., Capps, S.L., Hite, J.R., Jr, Carlton, A.G., Lee, S.H., Bergin, M.H., Ng, N.L.: Fine-particle water and pH in the southeastern United States, Atmos. Chem. Phys., 15, 5211-5228, 2015.

Hu, W., Hu, M., Hu, W., Jimenez, J. L., Yuan, B., Chen, W., Wang, M., Wu, Y., Chen, C., Wang, Z., Peng, J., Zeng, L., and Shao, M.: Chemical composition, sources, and aging process of submicron aerosols in Beijing: Contrast between summer and winter, J. Geophys. Res. Atmos., 121(4), 1955–1977, https://doi.org/10.1002/2015JD024020, 2016.

Hu, W., Hu, M., Hu, W.-W., Zheng, J., Chen, C., Wu, Y., and Guo, S.: Seasonal variations in high time-resolved chemical compositions, sources, and evolution of atmospheric submicron aerosols in the megacity Beijing, Atmos. Chem. Phys., 17, 9979–10000, https://doi.org/10.5194/acp-17-9979-2017, 2017.

Li, H., Zhang, Q., Zhang, Q., Chen, C., Wang, L., Wei, Z., Zhou, S., Parworth, C., Zheng, B., Canonaco, F., Prévôt, A. S. H., Chen, P., Zhang, H., Wallington, T. J., and He, K.: Wintertime aerosol chemistry and haze evolution in an extremely polluted city of the North China Plain: significant contribution from coal and biomass combustion, Atmos. Chem. Phys., 17, 4751–4768, https://doi.org/10.5194/acp-17-4751-2017, 2017.

Sun, Y. L., Wang, Z. F., Du, W., Zhang, Q., Wang, Q. Q., Fu, P. Q., Pan, X., Li, J., Jayne, J., and Worsnop, D. R.: Long-term real-time measurements of aerosol particle composition in Beijing, China: seasonal variations, meteorological effects, and source analysis, Atmos. Chem. Phys., 15, 10149–10165, https://doi.org/10.5194/acp-15-10149-2015, 2015.

Xu, W. Q., Han, T. T., Du, W., Wang, Q. Q., Chen, C., Zhao, J., Zhang, Y. J., Li, J., Fu, P. Q., Wang, Z. F., Worsnop, D. R., and Sun, Y. L.: Effects of Aqueous-Phase and Photochemical Processing on Secondary Organic Aerosol Formation and Evolution in Beijing, China, Environ. Sci. Technol., 51(2), 762–770, https://doi.org/10.1021/acs.est.6b04498, 2017.